# Hydrogen Gas Sensing Properties of Mixed Copper–Titanium Oxide Thin Films

**DOI:** 10.3390/s23083822

**Published:** 2023-04-08

**Authors:** Ewa Mańkowska, Michał Mazur, Jarosław Domaradzki, Piotr Mazur, Małgorzata Kot, Jan Ingo Flege

**Affiliations:** 1Faculty of Electronic, Photonics and Microsystems, Wrocław University of Science and Technology, Janiszewskiego 11/17, 50-372 Wrocław, Poland; 2Institute of Experimental Physics, University of Wrocław, Max Born 9, 50-204 Wrocław, Poland; 3Applied Physics and Semiconductor Spectroscopy, Brandenburg University of Technology Cottbus-Senftenberg, Konrad-Zuse-Strasse 1, 03046 Cottbus, Germany

**Keywords:** mixed copper–titanium oxides, Cu_2_O, TiO_2_, (CuTi)Ox, hydrogen gas sensing, thin films, magnetron sputtering

## Abstract

Hydrogen is an efficient source of clean and environmentally friendly energy. However, because it is explosive at concentrations higher than 4%, safety issues are a great concern. As its applications are extended, the need for the production of reliable monitoring systems is urgent. In this work, mixed copper–titanium oxide ((CuTi)Ox) thin films with various copper concentrations (0–100 at.%), deposited by magnetron sputtering and annealed at 473 K, were investigated as a prospective hydrogen gas sensing material. Scanning electron microscopy was applied to determine the morphology of the thin films. Their structure and chemical composition were investigated by X-ray diffraction and X-ray photoelectron spectroscopy, respectively. The prepared films were nanocrystalline mixtures of metallic copper, cuprous oxide, and titanium anatase in the bulk, whereas at the surface only cupric oxide was found. In comparison to the literature, the (CuTi)Ox thin films already showed a sensor response to hydrogen at a relatively low operating temperature of 473 K without using any extra catalyst. The best sensor response and sensitivity to hydrogen gas were found in the mixed copper–titanium oxides containing similar atomic concentrations of both metals, i.e., 41/59 and 56/44 of Cu/Ti. Most probably, this effect is related to their similar morphology and to the simultaneous presence of Cu and Cu_2_O crystals in these mixed oxide films. In particular, the studies of surface oxidation state revealed that it was the same for all annealed films and consisted only of CuO. However, in view of their crystalline structure, they consisted of Cu and Cu_2_O nanocrystals in the thin film volume.

## 1. Introduction

Metal oxides are materials that can change their electrical resistance when exposed to particular gas molecules. This phenomenon was first shown in germanium in the early 1950s by Brattain and Bardeen [1]. In the early 1970s, a resistive gas sensor device based on tin dioxide (SnO_2_) was fabricated and patented by Taguchi [2]. Today, the production of gas sensor devices is a crucial task due to the growing demand for monitoring of harmful and flammable substances in the environment and industrial processes. Moreover, gas-sensitive thin films are used in healthcare, e.g., in exhaled air analysers [3,4] and in wearable electronics [5,6]. Hydrogen, for example, is considered to be a clean energy vector for the automotive industry. However, it is explosive in a wide range of concentrations (4–75%), ignites easily, and its large diffusion coefficient causes problems with storage. Therefore, its concentration must be constantly monitored, and new strategies are currently being developed to create reliable hydrogen sensors.

Different oxides can be used as chemical gas sensors. Copper oxides are well-known gas sensing materials for reducing (CO, H_2_, CH_4_, NH_3_) and oxidising (NO_2_, SO_3_) gases, exhibiting a hole-type electrical conduction. In numerous works, the gas sensing performance of CuO [7,8,9,10], Cu_2_O [11,12], and their mixture [13,14,15] has been investigated. Titanium dioxide is another well-established gas sensing material that exhibits electron-type conduction [16,17,18,19]. A fairly common procedure, often used in chemical sensor technology, is to apply an additional, thin layer of a catalyst (Au, Pt, or Pd) that determines the gas sensing effect itself [20,21]. Moreover, a well-known method used to improve the performance of gas sensors is the application of mixed oxides such as WO_3_-SnO_2_ [22], ZnO-NiO [23], ZnO-CuO [24], ZrO_2_-Y_2_O_3_ [25], and MnCo_2_O_4_ [26].

Despite this, little is known about the gas sensing properties of mixed copper–titanium oxides. Such a mixture could be attractive compared to either copper oxides or titanium oxides alone thanks to a lower optimal operating temperature [27], which contributes to the reduction of sensor power consumption [28], greater long-term operating stability [28,29], and improved sensor response due to the depletion zone formed by the p–n heterojunction interface being more sensitive to the presence of gas than the depletion zone formed by oxygen adsorption/desorption [30]. Most previously published works concern double layers [6,31,32], nanowires or nanosheets decorated with nanoparticles [28,33,34], core–shell structure nanowires [35,36], or nanoparticles [37,38,39] of mixed copper and titanium oxides. Nakate et al. [38] compared the sensor responses of CuO/TiO_2_ nanoparticles to ethanol, hydrogen, and nitrogen oxides at a fixed concentration of 800 ppm at a working temperature of 523 K. The best response was ascribed to ethanol, then to hydrogen, and the worst performance was presented by NO_2_. Similarly, Lupan et al. [32] obtained an enhanced gas response to ethanol compared to hydrogen in the temperature range between 523 and 623 K, which was opposite the result of Barreca et al. [40], who showed that the response to hydrogen is about two times higher than that of ethanol. Furthermore, a mixed oxide of 50% TiO_2_ and 50% CuO was found to be more sensitive to NO_2_ than to H_2_, and could be further improved by adding Li to the mixture [41,42].

In this work, we study the influence of the morphology, crystal structure, and chemical composition of mixed copper–titanium oxide (CuTi)Ox thin films with various Cu concentrations along with their hydrogen sensing properties. To date, there have been a limited number of works on the hydrogen sensing properties of thin films based on mixed copper–titanium oxides, particularly those prepared by PVD methods such as magnetron sputtering. Using magnetron sputtering, it was possible to deposit coatings with homogenously distributed copper and titanium oxides in the volume of the thin films. These films had a higher response to hydrogen than the reference Cu_x_O and TiO_x_ films. Sensing responses comparable to those shown in the literature were obtained for (CuTi)Ox films, which consisted of Cu-Cu_2_O or Cu_2_O crystallites (depending on the amount of Cu in the thin film), whereas CuO was only present on the surface as a native oxide. Such a crystal structure consisting of mixed Cu-Cu_2_O or Cu_2_O has never before been observed in studies of the sensing properties of copper–titanium oxides. Furthermore, despite the absence of a catalyst, (CuTi)Ox can operate at a relatively low temperature of 473 K, which is a novelty compared to the existing literature, in which such sensors typically work at temperatures of 573 K or higher.

## 2. Materials and Methods

The subjects of this study were nanostructured (CuTi)Ox thin films in which copper and titanium were not fully oxidized directly after the deposition process. A representative set of samples with various elemental compositions was fabricated using reactive magnetron sputtering. The copper content in the thin films was estimated to be in the range of 20 to 80 at.% using energy dispersive X-ray spectrometry. During the deposition process, MSS2 2 kW pulsed DC power supply units (DORA Power System, Denizli, Turkey) powered three magnetrons equipped with two circular titanium targets (99.999%) and one copper target (99.999%), each with a diameter of 28.5 mm and a thickness of 3 mm. Various material compositions were achieved by changing the average power released to each magnetron using the pulse width modulation method, that is, the *pwm* coefficient. The base pressure was at the order of 10^−5^ mbar, while the pressure in the vacuum chamber during the sputtering was kept at approximately 10^−2^ mbar. Detailed deposition parameters were as described in our previous article [43]. The thin films were deposited on fused silica (SiO_2_), silicon substrates, and ceramic substrates with interdigitated platinum–gold electrodes, and after deposition were annealed at 473 K in an air atmosphere in a tubular furnace (Nabertherm RS (80/300/11)) equipped with a quartz tube.

The morphology measurements were performed with the aid of an FEI Helios Xe-PFIB field-emission scanning electron microscope (FE-SEM) equipped with an EDS detector (Bruker). The atomic content (%Cu/(%Cu + %Ti)) of copper (%Cu) in relation to titanium (%Ti) in the thin film oxide mixtures was estimated to be 23 at.%, 41 at.%, 56 at.%, and 77 at.% [43]. The thickness of the as-deposited thin films was measured using a Taylor Hobson (Talysurf CCI Lite) optical profilometer, and was equal to 540, 620, 670, 580, 430, and 430 nm for TiO_x_, (Cu_0.23_Ti_0.77_)Ox, (Cu_0.41_Ti_0.59_)Ox, (Cu_0.56_Ti_0.44_)Ox, (Cu_0.77_Ti_0.23_)Ox, and Cu_x_O, respectively.

To determine the microstructure, X-ray diffraction (XRD) measurements were performed in a grazing incidence mode (GIXRD) using an Empyrean (PANalytical) X-ray diffractometer with a PIXel3D detector (a Cu Kα X-ray source with a wavelength of 1.5406 Å).

With the aid of MDI JADE 5.0 software, the diffraction patterns were collected with a step equal to 0.05° in the 2θ range of 30° to 80° and subsequently compared with PDF cards (Cu #04-0836 [44], Cu_2_O #65-3288 [45], CuO #65-2309 [46], TiO #65-2900 [47], and anatase #21-1272 [48]).

Furthermore, the oxidation states of copper and titanium located near the surface (up to approximately 10 nm below the sample surface) were investigated by X-ray photoelectron spectroscopy (XPS) using a Specs Phoibos 100 MCD-5 (5-channel electron multiplier) hemispherical analyzer. Specs XR-50 and SPECS HSA3500 were used as the X-ray sources of non-monochromatic Al Kα (1486.6 eV) radiation.

The gas sensing properties of the annealed (CuTi)Ox thin films were measured in a sealed Instec chamber equipped with a hot chuck controlled by an Instec mK1000 temperature controller at a working temperature of 473 K. Before exposure to the target gas, the samples were stabilized at 473 K in an ambient air environment for an hour. Ambient air flow was provided by an Atlas Copco GX3 FF compressor with integrated refrigerant air drier that provided a humidity level of <10% and an activated carbon adsorbing filter (Donaldson AK series). In the case of the target gases, their various concentrations were obtained by modification of the gas flow rate ratio of the target gas and an inert gas (i.e., argon) using a dual-channel MKS PR4000B Digital Power Supply and Readout, two MKS mass flow controllers, and Swagelok valves. The gas sensing properties were measured for hydrogen concentrations of 100, 200, 500, and 1000 ppm. Sensor resistance measurements were performed using an Agilent 34901A data acquisition system connected through a GPIB interface to a PC with control software written in the TestPoint environment.

## 3. Results and Discussion

Figure 1 shows images of the as-deposited and annealed at 473 K (CuTi)Ox thin films taken using a scanning electron microscopy (SEM) method. Post-deposition annealing at 473 K did not affect the morphology of the reference TiO_x_ thin film, as before and after annealing it consisted of densely packed columnar grains. Mixed copper–titanium oxides with less than 60 at.% of copper consisted of densely packed elongated grains, while after annealing their structure changed throughout the depth of the sample. Near the substrate, the grains retained their elongated character, a granular structure appeared in the near-surface layer, and the depth of the morphology modification was dependent on the amount of copper in the thin film. When further increasing the Cu concentration to 77 at.%, the as-deposited densely packed (Cu_0.77_Ti_0.23_)Ox film changed to a structure of elongated grains with spaces between them. Certain grains were smaller at the bottom, and in general the grains widened in an upwards direction. As a consequence, the inter-grain voids appeared, and after annealing they were transformed into pores. The as-deposited reference Cu_x_O thin film consisted of fibrous grains, and individual voids were observed. As a result of annealing, uniformly distributed pores appeared in the entire structure. In summary, in all of the copper-containing thin films the post-deposition annealing process changed the relatively flat surfaces to 3D grains.

Figure 2 presents the results of the XRD experiments. The diffractograms of the as-deposited and annealed at 473 K TiO_x_ films consist of two peaks centered at 43.74° and 63.52°, which can be attributed to the (200) and (220) reflections of TiO. Scherrer analysis revealed that titanium monoxide crystallites were smaller than 10 nm. As-deposited thin films with copper amount between 23 and 56 at.% were composed only of crystalline metallic copper, while after annealing additional reflections from Cu_2_O crystal were observed. Furthermore, in the spectra of annealed (Cu_0.23_Ti_0.77_)Ox and (Cu_0.41_Ti_0.59_)Ox a peak attributed to the anatase TiO_2_ (101) crystal plane was found at 2θ equal to 25.28°. The sizes of the anatase crystallites were equal to 16 nm and 20 nm, respectively. Next, the as-deposited (Cu_0.77_Ti_0.23_)Ox and Cu_x_O consisted of the beginning of the crystalline phases of Cu, Cu_2_O, and CuO, while after annealing these changed mainly to crystal cuprous oxides. For these annealed thin films, the size of the crystallites was less than 20 nm. A thorough discussion of the crystalline structure of the as-deposited and annealed at 473 K (CuTi)Ox thin films can be found elsewhere [43].

XPS measurements were performed to determine the surface chemical composition of the prepared oxides before and after annealing. In the survey scans (Appendix A), in addition to components related to Cu, Ti, and O, other elements, e.g., carbon were found as well. A detailed analysis is provided in the Appendix A for this article. The analysis of the O1s core level is presented in Appendix A. After post-process annealing, the majority components in the thin films can be attributed to the lattice of copper oxide in the case of (CuTi)Ox and the lattice of TiO_2_ in the case of TiO_x_. For thin films annealed at 473 K, corrections for sample charging, if any, were made by calibrating the main peak in the C1s core level spectra to 284.8 eV [49]. All analyzed core levels were fitted with fixed values for the intensity ratio and separation (±0.1 eV) of the spin–orbit split peaks as well as the peak widths, i.e., the full width at half maximum (FWHM).

Our XRD investigations showed that TiO_x_ thin films consist of TiO crystallites. The position of the Ti2p_3/2_ peak for TiO is equal to 455.5 eV [50,51]; however, for our TiO_x_ thin films (Figure 3a) the position was in the range of 458.1–458.7 eV and the separation energies between Ti2p_1/2_ and the Ti2p_3/2_ doublets was ca. 5.8 eV. This indicates that, due to the exposure of the as-deposited and annealed TiO_x_ samples to air, the surface became oxidized because of the well-known self-oxidation effect of titanium. Furthermore, titanium dioxide was found in the Ti2p XPS spectra on the surface of the as-deposited (Cu_0.77_Ti_0.23_)Ox thin film; this vanished after annealing. Titanium oxides were not found in the Ti2p XPS spectra of the other (CuTi)Ox surfaces, possibly due to surface contamination that covered this signal and due to the migration of copper ions to the surface, which is another well-known effect [52].

According to the literature, the Cu2p core level is located in the binding energy range of 965 to 925 eV. In this region, peaks attributed to Cu2p_1/2_ and Cu2p_3/2_ spin–orbit coupling are found. Their separation energy is around 19.6 eV, while the intensity ratio should be equal to 1:2. For cupric oxide, the Cu2p_3/2_ peak is located at 934.0 eV [53]; additionally, at higher binding energy (~7.7 and 10.1 eV), shake-up satellite peaks are observed [54], which occur because the kinetic energy of the outgoing photoelectron becomes reduced owing to a further electronic excitation of the ion from its ground state. According to the literature, the satellite peak positions originating from the Cu^0^ and Cu^1+^ states are equal to 932.61 eV and 932.43 eV [50], respectively; thus, their difference is below the resolution limit of lab-based XPS. The FWHM of the 0 and 1+ oxidation states in the Cu2p core level is much smaller than the FWHM of the Cu^2+^ oxidation state. Therefore, to determine the chemical composition of the sample from the Cu2p core level analysis, a common procedure is to calculate a modified Auger parameter. This is defined as the sum of the binding energy and the kinetic energy of the Auger transition. Its advantage over the Auger parameter is the fact that it is independent of the excitation energy [50].

The analysis of the Cu2p core level spectra of the as-prepared samples (Figure 3b) was carried out by subtracting an inelastic background using Shirley’s method, which is one of the three commonly used methods [55]. In the fitting procedure, the constraint of the peak shape was fixed to be 70% Gaussian and 30% Lorentzian in agreement with [50,56]. The peak in the Cu2p_3/2_ area was deconvoluted into two peaks positioned at about 932.3 and 934.0 eV. The peaks located at higher binding energies, along with satellite peaks separated from the line at 934.0 eV + 7.7 and +10.1 eV, indicate the presence of CuO on the surface of the thin films. It should be noted that in the Cu2p^2+^ satellite region the satellites from the Cu2p_1/2_ line appear as a consequence of the usage of the non-monochromatized Al Kα source. The modified Auger parameter calculated for the as-deposited thin films was in the range of 1849.0 to 1849.5 eV, indicating the presence of copper(I) oxide (Cu_2_O) instead of metallic copper, for which the literature value of the modified Auger parameter is equal to 1851.24 eV [50]. Furthermore, the amount of CuO in the mixed copper oxides was determined using Equation (1):(1)$${\mathrm{Cu}}^{2+}=\frac{{\mathrm{Cu}}_{\mathrm{II}}+S}{{\mathrm{Cu}}_{I}+{\mathrm{Cu}}_{\mathrm{II}}+S}\cdot 100\%$$
where Cu_I_—area under peak Cu^+^ 2p_3/2_, Cu_II_—area under peak Cu^2+^ 2p_3/2_, S—area under the satellite peak.

The smallest amounts (~30%) of CuO were found on the surface of the reference Cu_x_O thin film and on (Cu_0.41_Ti_0.59_)Ox. For the other thin films, the content was greater than 55%. These results are in agreement with the intensity of Cu2p_3/2_ of CuO and the shake-up satellite.

By comparing the shape of the Cu LMM peak and its position with data from the literature [56] and conducting the analysis of the Cu2p region, it was found that in the near-surface layer the post-deposition annealing process resulted in the complete oxidation of copper(I) to copper(II) oxide. A summary of the XPS results is shown in Table 1.

Based on the XPS results analysis of the as-prepared and annealed metal oxide films, it can be concluded that even if the as-prepared and stored for a long time metal oxides surfaces are significantly different the annealing process for four hours at 473 K removes all of the adsorbed contamination from their surfaces and fully oxidizes the metal oxide surfaces, causing them to become spectroscopically uniform at a depth of about 10 nm.

The differences in the copper and titanium oxidation states obtained from the XRD and XPS investigations can be explained, e.g., by their different information depth. As mentioned above, XPS is a surface-sensitive method (information depth up to about 10 nm below sample surface), while XRD is more bulk-sensitive and reflects only the crystal phases of the film in its patterns. The amorphous phases cannot be investigated by XRD, even if they exist in the bulk of the film. Therefore, by combining those two methods it is possible to learn about the surface and bulk properties of the investigated films. Titanium that is exposed to ambient air is easily oxidized even at room temperature. This process is called passivation, and is the reason for the high corrosion resistance of titanium [57]. This phenomenon explains the fact that in this study on TiO_x_ thin films a crystalline TiO was found in the bulk, while TiO_2_ was found on the surface. Similarly, native copper oxides are known to form on the surface of metallic copper at room temperature. Furthermore, the presence of CuO depends on the crystalline structure of the surface of Cu and its morphology [58,59]. In this work, the thin films in the bulk consisted of Cu/Cu_2_O or only Cu_2_O crystallites. The samples were stored under atmospheric air conditions, which resulted in the self-oxidation effect and the formation of Cu_2_O/CuO mixtures at the surface of the as-deposited thin films and CuO layer at the top of the annealed (CuTi)Ox.

In the next step, we checked the response of the (CuTi)Ox films to hydrogen gas. Figure 4a shows the changes in the electrical resistance of the TiO_x_, Cu_x_O, and mixed copper–titanium oxide thin films annealed at 473 K upon exposure to 100–1000 ppm of hydrogen as a function of time. These results can be compared with the morphology, crystallinity, and surface chemistry of the annealed samples presented above.

Pure titanium oxide reacted with hydrogen as a typical n-type metal oxide, as its resistance decreases when hydrogen is introduced into the chamber. Next, in contrast to the TiO_x_ film, all thin films containing copper behaved as typical p-type gas sensors, that is, their resistance increased in the presence of hydrogen [60].

The dynamics of the resistance changes in gas sensors are characterized by the so-called response and recovery times, which are defined as the time required to reach 90% of the difference between R_air_ and R_gas_ after introducing hydrogen and air, respectively [6]. In this work, each oxide film was first exposed to hydrogen for one hour and then to air for another hour. The response time for all samples exceeded 40 min, while the recovery time was significantly shorter and lasted between 5 and 37 min. For our samples, a general trend can be observed that the recovery time increased independently of the hydrogen concentration as the copper concentration in the thin film changed in the following order: 56, 0, 23, 41, 77, and 100 at.%.

To define the quality of the sensing properties of the investigated thin films, the sensor response (SR) and sensitivity (S) were determined. The sensor response was defined for p-type Cu_x_O and (CuTi)Ox materials according to the following equation [61]:(2)$$\mathrm{SR}=\left(\frac{R_{H_{2}}}{R_{\mathrm{air}}}\right)$$
and for the n-type TiO_x_ thin film according to the equation
(3)$$\mathrm{SR}=\left(\frac{R_{\mathrm{air}}}{R_{H_{2}}}\right)$$
where R_H_2__ is the resistance of thin film in a hydrogen atmosphere and R_air_ is the thin film resistance in ambient air.

When comparing the sensor responses of mixed copper–titanium oxides with pure Cu_x_O and TiO_x_, an increase in the SR values for (CuTi)Ox with a copper concentration exceeding 40 at.% was observed. The best sensor response to 100 ppm H_2_ was found for (Cu_0.41_Ti_0.59_)Ox, and was equal to 1.38. For higher concentrations of hydrogen, the best sensing behaviour was observed for (Cu_0.56_Ti_0.44_)Ox, then for (Cu_0.41_Ti_0.59_)Ox, and finally for (Cu_0.77_Ti_0.23_)Ox. XPS analysis showed that, independently of the elemental composition of the thin films, cupric oxide covered the surface of the mixed oxide thin films annealed at 473 K. Therefore, it can be assumed that the highest SR values observed for the thin films with 56 and 41 at.% of Cu is not related to the surface properties of the films, and is rather related to their crystalline structure and morphology. We have previously presented evidence that the crystalline mixture of metallic Cu and Cu_2_O affects the response of Cu_x_O thin films to hydrogen more strongly than the growth of CuO nanowires on the surface of the film [9]. When the inert metallic core–metal oxide structure is formed, the electronic properties (carrier distribution) of, e.g., Cu_2_O are modified, improving the SR [62,63]. Furthermore, the finding of a slightly higher SR of (Cu_0.56_Ti_0.44_)Ox compared to (Cu_0.41_Ti_0.59_)Ox might be related to its larger active surface area (Figure 1). On the other hand, Cu_x_O and (Cu_0.77_Ti_0.23_)Ox are both mainly composed of Cu_2_O crystallites, their thickness is equal to 430 nm, and their morphology is mostly porous. However, the copper–titanium oxide mixture exhibits higher SR values than the reference sample, and it can be assumed that the amorphous titanium oxide improves the hydrogen gas sensing performance of copper oxide-based thin films.

Next, sensitivity (S) was defined for all films as the slope of the relationship between SR and H_2_ concentration. The Pearson’s correlation coefficients were greater than 0.95, except (Cu_0.23_Ti_0.77_)Ox, for which it was 0.60, proving that the SR increases linearly with increasing hydrogen concentration. The highest sensitivity, equal to 5.4·10^−4^ (unit per 1 ppm of H_2_), was exhibited by the (Cu_0.56_Ti_0.44_)Ox thin film. The extrapolated SR values based on the sensitivity are in agreement with the experimental data for the H_2_ concentration equal to 35,000 ppm presented in [43]. An exception was observed for the (CuTi)Ox thin film with 77 at.% of copper, for which the experimental SR was more than 4.5 times higher than the extrapolated value.

In addition, we performed repeatability tests. The performance of the thin films was repeatable, and the small deviations from the sensor response were mainly caused by a slight drift in the baseline. Figure 5 shows the resistance changes in 200 ppm hydrogen and an atmosphere of ambient air for the (Cu_0.77_Ti_0.23_)Ox thin film. The mean SR value was equal to 1.35, and the standard deviation was 0.04.

Table 2 summarizes the hydrogen gas sensing parameters of the copper and titanium oxide nanomaterials available in the literature. To date, hydrogen-sensitive structures operating at 473 K have been mainly manufactured as double layers [32] in which, for example, TiO_x_ is in the form of nanotubes [35]. To obtain a nanocomposite with dispersed oxides, an additional thermal treatment can be used [40]. Similar to this work, the hydrogen gas concentration was in the range between 100 and 1000 ppm. In general, in all these works the SR does not exceed 2.0 at an operating temperature of 473 K, while the response of the sensor has been determined by such a variety of methods that it is difficult to compare the individual results with each other. However, based on the literature and the results presented in this study, it seems worthwhile to carry out an in-depth study on the sensing properties of (CuTi)Ox thin films. It is worth mentioning that the results presented in this work, in comparison to those presented in the literature, showed thin oxide films with different crystal structures. The phenomena of gas sensing in copper–titanium oxides have mostly been investigated in terms of their structures, which for copper oxides mostly refers to crystalline structure of pure CuO [41] or Cu_2_O-CuO structures [32,35,40,64]. In the case of this work, the copper oxides were composed of the nanocrystallites of Cu-Cu_2_O or Cu_2_O, while CuO was only observed on the surface. To the best of our knowledge, such thin films incorporating mixed oxides of copper at various oxidation states with titanium have not previously been reported.

## 4. Conclusions

SEM, XRD, XPS and electrical characterization studies were conducted for four magnetron sputtered mixed copper–titanium oxides and two reference samples of pure Cu_x_O and TiO_x._ These measurements showed that their response and sensitivity in detecting hydrogen gas depended more on their morphology and crystallinity than on their surface chemistry, as the former, according to XPS studies, seems to be the same for all Cu-containing samples. Cross-sectional SEM images revealed that significantly increasing the atomic concentration of copper in the Cu-Ti mixed oxides leads to the films becoming rougher and more porous after annealing, thereby increasing the active surface area available for hydrogen gas sensing, which is reflected in improved SR and S parameters when increased Cu content is present in the mixed films. XRD analysis of the annealed samples showed that the reference TiO_x_ sample was partially crystalline, which took the form of TiO. By adding 23 at.% of Cu to the Cu-Ti mixed oxide during magnetron sputtering and annealing it at 473 K, the resulted film shows the reflections from the anatise-TiO_2_ crystallites instead of TiO. In addition, in the sample with 41 at.% of Cu, the reflection from anatase–TiO_2_ is detected, though with smaller intensity, and completely disappears with further increase in the at.% of Cu in Cu-Ti mixed oxides. Moreover, we found a correlation between the improved Cu_2_O crystallinity and the increased at.% of Cu in the annealed mixed oxides. In addition, in the samples with 41, 56, and 77 at.% of Cu, a reflection from the metallic Cu was visible, while the sample with 56 at.% of Cu contained the most. The (CuTi)Ox thin films containing from 41 to 77 at.% of copper exhibited an enhanced sensor response compared to the reference Cu_x_O and TiO_x_ thin films. Therefore, we can assume that the presence of crystalline Cu and Cu_2_O at the same time in the copper–titanium mixed oxide films may be responsible for the improved sensor response. The highest sensor response and sensitivity along with the shortest recovery time was observed for the (Cu_0.56_Ti_0.44_)Ox and (Cu_0.41_Ti_0.59_)Ox films, which could be explained by the specific combination of morphology and crystal structure. These films were characterized by the highest active surface area and by a nanocrystalline structure composed of metallic Cu and Cu_2_O, while in both cases the surface was covered by CuO. As next steps, we plan to thoroughly investigate the hydrogen gas sensing properties of (CuTi)Ox with and without a catalyst at various hydrogen concentrations lower than those presented in the paper and at higher operating temperatures, as well as to further develop the hydrogen detection mechanism.

## Figures and Tables

**Figure 1 sensors-23-03822-f001:**
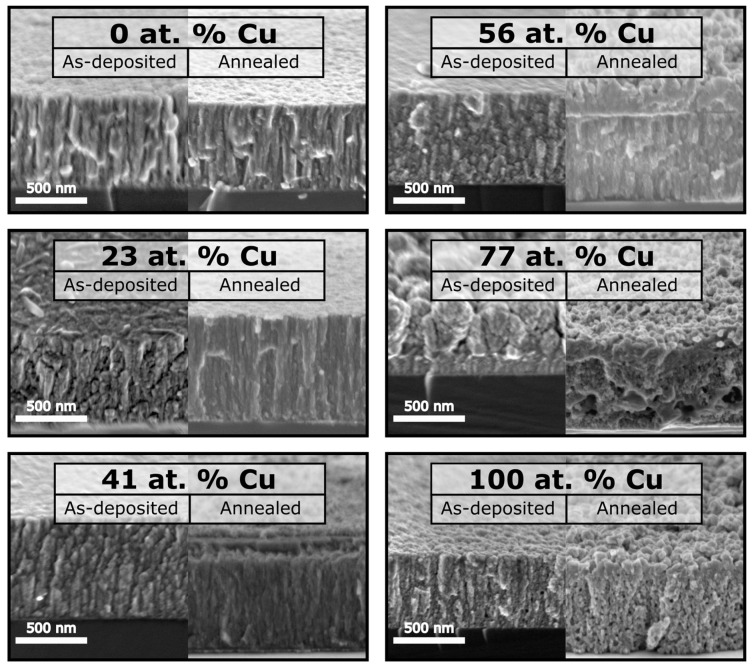
SEM cross-section images of as-deposited and annealed at 473 K TiO_x_, (CuTi)Ox, and Cu_x_O thin films.

**Figure 2 sensors-23-03822-f002:**
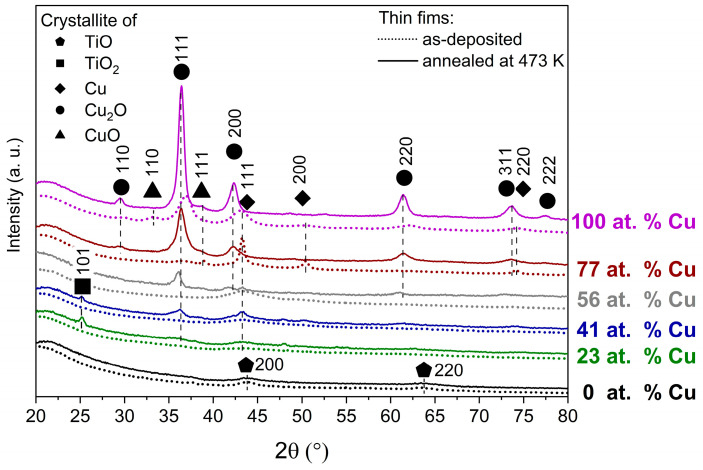
The diffraction patterns of the as-deposited and annealed at 473 K TiO_x_, (CuTi)Ox, and Cu_x_O thin films.

**Figure 3 sensors-23-03822-f003:**
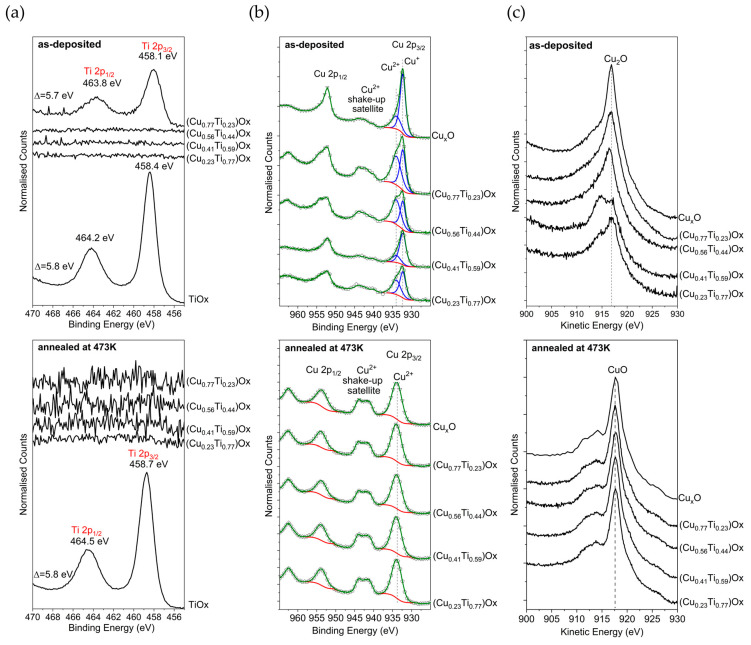
XPS spectra of (**a**) Ti2p; (**b**) Cu2p; and (**c**) Cu LMM of the as-deposited and annealed thin films.

**Figure 4 sensors-23-03822-f004:**
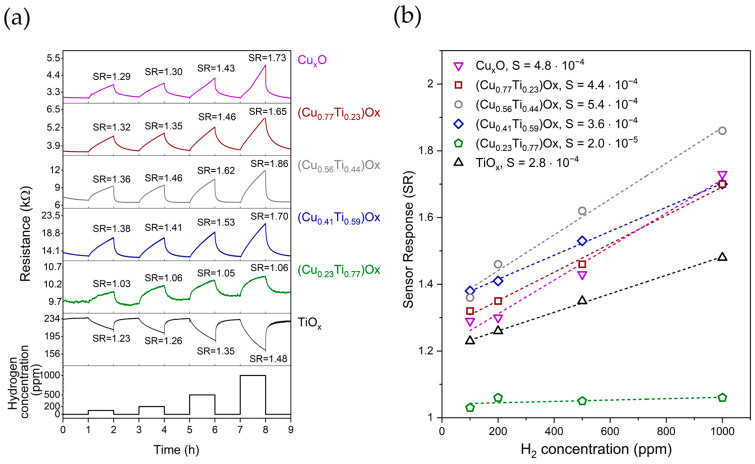
(**a**) Changes in electrical resistance of TiO_x_, (CuTi)Ox, and Cu_x_O thin films annealed at 473 K upon exposure to 100–1000 ppm of H_2_; (**b**) the sensor response as a function of hydrogen concentration.

**Figure 5 sensors-23-03822-f005:**
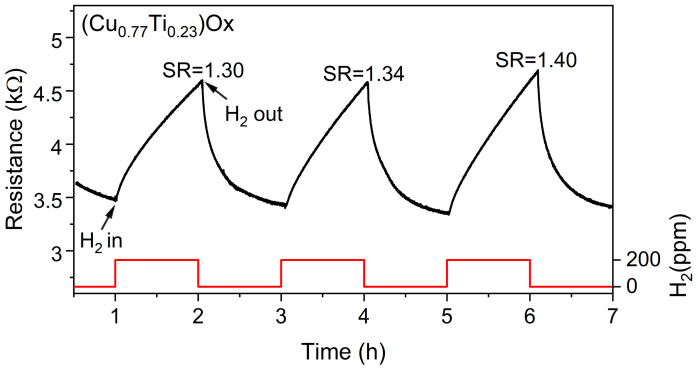
Three cycles of changes in electrical resistance of (Cu_0.77_Ti_0.23_)Ox thin film annealed at 473K upon exposure to 200 ppm of H_2_.

**Table 1 sensors-23-03822-t001:** Summary of Cu2p core level peak positions, modified Auger parameters, and CuO concentration of as-deposited thin films.

As-Deposited Thin Film	Cu 2p_3/2_		Modified Auger Parameter (eV)	CuO Content in the Mixture of Copper Oxides
	Cu^+^ (eV)	Cu^2+^ (eV)		
(Cu_0.23_Ti_0.77_)Ox	932.3	934.0	1849.1	55%
(Cu_0.41_Ti_0.59_)Ox	932.3	934.0	1849.5	31%
(Cu_0.56_Ti_0.44_)Ox	932.3	934.0	1849.0	68%
(Cu_0.77_Ti_0.23_)Ox	932.3	934.0	1849.2	64%
Cu_x_O	932.3	934.0	1849.1	34%

**Table 2 sensors-23-03822-t002:** Summary of sensor responses of nanomaterials based on copper and titanium oxides measured at a working temperature of 473 K.

Heterostructure	Hydrogen Concentration (ppm)	Sensor Response at 473 K	Equation of SR	Reference
(CuTi)O_x_ thin films	1000	1.86	R_gas_/R_air_	This work
CuO-TiO_2_ nanocomposite	1000	2.0	R_gas_/R_air_	[40]
CuO thin film on TiO_2_ nanotube	1000	2.0	(I_gas_ − I_air_)/I_air_	[35]
CuO-TiO_2_ nanocages	800	150% (at 523 K)	(R_gas_ − R_air_)/R_air_	[38]
TiO_2_/Cu_x_O	100	10%	(R_gas_ − R_air_)/R_air_	[32]
TiO_2_/Cu_x_O	100	43%	R_air_/R_gas_	[64]

## Data Availability

Not applicable.

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
