# Peer review of "Hydrogen Gas Sensing Properties of Mixed Copper–Titanium Oxide Thin Films"

_sensors, 2023, doi:10.3390/s23083822_

Round 1
Reviewer 1 Report
The mixed copper-titanium oxide ((CuTi)Ox) thin films were fabricated around 500 nm with various copper concentrations, which were investigated as hydrogen gas sensor in this manuscript. The work is worthy to be accepted, but the following issues should be considered before acceptance.
1, In the introduction, it is better to introduce the potential advantages of such ((CuTi)Ox) thin film for hydrogen detection compared with other nanostructured materials, such as CuO/TiO2 nanoparticles.
2, In Figure 2, it is hard to comprehend why the TiO or TiO2 peak is so weak when the Cu concentration is 0%? The intensity of which is weaker than the sample with lower Ti concentration.
3, There exist some mistakes in writing, such as the symbol "÷ " is often used to represent "to" or "-", which will confuse readers.
4, The references should be cited if the data is not obtained from this manuscript, for example, line 183, "the position of the peak originating 183
from the Cu0 state is equal to 932.61 eV, and from the Cu+1 state it is equal to 932.43 eV".
.
Author Response
The detailed answers are in the attachment.
Answers to the report of Reviewer #1
on the manuscript entitled: “Hydrogen gas sensing properties of mixed copper-titanium oxides thin films”
Authors: Ewa Mańkowska, Michał Mazur, Jarosław Domaradzki, Piotr Mazur, Małgorzata Kot, Jan Ingo Flege
Authors:
We would like to express our gratitude for your remarks, which let us improve our manuscript. We have taken them into account in the revised version of our paper.
Answering to the Reviewer’s remarks, we have introduced some revisions in the manuscript.
Reviewer:
In the introduction, it is better to introduce the potential advantages of such ((CuTi)Ox) thin film for hydrogen detection compared with other nanostructured materials, such as CuO/TiO2 nanoparticles.
Authors:
A short comparison of (CuTi)Ox thin films presented in the paper with other structures published so far was added in the discussion of obtained results.
“Table 2 summarizes the hydrogen gas sensing parameters of copper and titanium oxide nanomaterials available in the literature. To date, hydrogen-sensitive structures operating at 473 K have been mainly manufactured as double layers [32] in which, for example, TiOx was in the form of nanotubes [35]. To obtain a nanocomposite with dispersed oxides, an additional thermal treatment was used [40]. Similarly to this work, the hydrogen gas concentration was in the range between 100 and 1000 ppm. In general, in all these works, the SR does not exceed 2.0 at an operating temperature of 473 K, but the response of the sensor was determined by such a variety of methods that it is difficult to compare individual results with each other. However, based on the literature and the results presented in this study, it seems worth-while to carry out an in-depth study on the sensing properties of (CuTi)Ox thin films. It is worth mentioning that the results presented in this work, in comparison to those presented in the literature, showed thin oxide films with different crystal structure. The phenomena of gas sensing in copper-titanium oxides are mostly investigated in the structures in which copper oxides mostly refers to crystalline structure of pure CuO [41] or Cu2O-CuO structures [32,35,40,64]. In the case of this work, the copper oxides are composed of the nanocrystallites of Cu-Cu2O or Cu2O, while CuO is only observed on the surface. According to the authors best knowledge, such thin films of mixed oxides of copper at various oxidation states with titanium have not been reported to date.”
Table 2. The summary of sensors responses of nanomaterials based on copper and titanium oxides measured at the working temperature of 473 K
|
Heterostructure |
Hydrogen concentration (ppm) |
Sensor Response at 473 K |
Equation of SR |
Reference |
|
(CuTi)Ox thin films |
1000 ppm |
1.86 |
Rgas/Rair |
This work |
|
CuO-TiO2 |
1000 |
2.0 |
Rgas/Rair |
[40] |
|
CuO thin film on TiO2 nanotube |
1000 |
2.0 |
(Igas-Iair)/Iair |
[35] |
|
CuO-TiO2 |
800 |
150% |
(Rgas-Rair)/Rair |
[38] |
|
TiO2/CuxO |
100 |
10% |
(Rgas-Rair)/Rair |
[32] |
|
TiO2/CuxO |
100 |
43% |
Rair/Rgas |
[63] |
Reviewer:
In Figure 2, it is hard to comprehend why the TiO or TiO2 peak is so weak when the Cu concentration is 0%? The intensity of which is weaker than the sample with lower Ti concentration.
Authors:
TiOx thin films deposited on the substrate being at room temperature are mostly amorphous in nature [1,2]. Crystallisation of TiO2 is observed at temperatures exceeding 673 K, which is significantly higher than the temperature of annealing presented in our work (i.e. 473 K). The impact of copper additive to TiOx on the crystallisation temperature still needs to be developed, however, there are reports indicating that in a certain range of concentrations, copper may affect the crystallization of anatase, i.e. decreasing its crystallization temperature [3,4].
- Ambardekar, V.; Bhowmick, T.; Bandyopadhyay, P.P. Understanding on the Hydrogen Detection of Plasma Sprayed Tin Oxide/Tungsten Oxide (SnO2/WO3) Sensor. Int. J. Hydrogen Energy 2022, 47, 15120–15131, doi:10.1016/j.ijhydene.2022.03.005.
- Suhail, M.H.; Rao, G.M.; Mohan, S. Dc Reactive Magnetron Sputtering of Titanium-Structural and Optical Characterization of TiO2 Films. J. Appl. Phys. 1992, 71, 1421–1427, doi:10.1063/1.351264.
- Yang, C.; Hirose, Y.; Nakao, S.; Hasegawa, T. TiO2 Thin Film Crystallization Temperature Lowered by Cu-Induced Solid Phase Crystallization. Thin Solid Films 2014, 553, 17–20, doi:10.1016/j.tsf.2013.12.041.
- Baltazar, P.; Lara, V.H.; Córdoba, G.; Arroyo, R. Kinetics of the Amorphous - Anatase Phase Transformation in Copper Doped Titanium Oxide. J. Sol-Gel Sci. Technol. 2006, 37, 129–133, doi:10.1007/s10971-006-6432-0.
Reviewer:
There exist some mistakes in writing, such as the symbol "÷ " is often used to represent "to" or "-", which will confuse readers.
Authors:
The symbol "÷ "was replaced with "-".
Reviewer:
The references should be cited if the data is not obtained from this manuscript, for example, line 183, "the position of the peak originating 183from the Cu0 state is equal to 932.61 eV, and from the Cu+1 state it is equal to 932.43 eV"
Authors:
The references were added to the manuscript.

Reviewer 2 Report
This manuscript studied mixed copper-titanium oxide thin films with various copper concentrations and their application as H2 sensor. After carefully read this manuscript, I notice two major weaknesses. First, the authors used many paragraphs to discuss and analyze the XPS results. They eventually obtained the component ratio of CuO in all samples. However this to me is meaningless as the ratio of CuO is not only affected by the at. % ratio of Cu and Ti, but also many other factors, including the concentration of Oxygen vacancy in the oxide film, annealing temperature, etc. Second, the authors compared the SR to H2 of samples with different Cu concentrations but didn’t provide insight into the reason that causes the SR difference. Therefore I would suggest that the authors take a systematic study to draw a solid conclusion and resubmit their work.
In addition to these, several other points also need to be addressed and clarified:
1. How were the samples wired for H2 sensing measurement?
2. Miller indices are supposed to be labeled when doing XRD analysis.
3. In Fig 5a, Why is the resistance of TiOx much higher than the rest of samples?
4 . The authors mentioned that adding an additional layer of catalyst is a common procedure for making a gas sensor. Does the samples tested in this manuscript have a catalyst layer?
Author Response
The detailed responses are in the attachment (together with figures).
Answers to the report of Reviewer #2
on the manuscript entitled: “Hydrogen gas sensing properties of mixed copper-titanium oxides thin films”
Authors: Ewa Mańkowska, Michał Mazur, Jarosław Domaradzki, Piotr Mazur, Małgorzata Kot, Jan Ingo Flege
Authors:
We would like to express our gratitude for your remarks, which let us improve our manuscript. We have taken them into account in the revised version of our paper.
Answering to the Reviewer’s remarks, we have introduced some revisions in the manuscript.
Reviewer:
This manuscript studied mixed copper-titanium oxide thin films with various copper concentrations and their application as H2 sensor. After carefully read this manuscript, I notice two major weaknesses. First, the authors used many paragraphs to discuss and analyze the XPS results. They eventually obtained the component ratio of CuO in all samples. However this to me is meaningless as the ratio of CuO is not only affected by the at. % ratio of Cu and Ti, but also many other factors, including the concentration of Oxygen vacancy in the oxide film, annealing temperature, etc.
Authors:
One of the aims of this work is to show how the structure and morphology of (CuTi)Ox thin films changed after post-deposition annealing process. To this end, the results of XRD, XPS and SEM images were analyzed. Authors agree that CuO content at the surface is not only affected by Cu and Ti content, however it is a common procedure to specify the chemical composition of the surface before and after annealing.
Reviewer:
Second, the authors compared the SR to H2 of samples with different Cu concentrations but didn’t provide insight into the reason that causes the SR difference. Therefore I would suggest that the authors take a systematic study to draw a solid conclusion and resubmit their work.
Authors:
A discussion of the SR values was extended to:
“When comparing the sensor responses of mixed copper-titanium oxides with pure CuxO and TiOx, an increase in the SR values for (CuTi)Ox with a copper concentration exceeding 40 at. % can be observed. The best sensor response to 100 ppm H2 was found for (Cu0.41Ti0.59)Ox and was equal to 1.38. For higher concentrations of hydrogen, the best sensing behaviour was observed for (Cu0.56Ti0.44)Ox, and then for (Cu0.41Ti0.59)Ox, and finally for (Cu0.77Ti0.23)Ox. XPS analysis showed that independently of the elemental composition of the thin films, cupric oxide covered the surface of the mixed oxide thin films annealed at 473 K. Therefore, it can be assumed that the highest SR values observed for the thin films with 56 and 41 at. % of Cu is not related to the surface properties of the films but rather to their crystalline structure and morphology. We have previously presented evidence that the crystalline mixture of metallic Cu and Cu2O affects the response of CuxO thin films to hydrogen more strongly than the growth of CuO nanowires on the surface of the film [9]. When the inert metallic core – metal oxide structure is formed, the electronic properties (carrier distribution) of, e.g., Cu2O are modified, improving the SR [62,63]. Furthermore, the finding of a slightly higher SR of (Cu0.56Ti0.44)Ox compared to (Cu0.41Ti0.59)Ox might be related to a larger active surface area (Figure 1). On the other hand, CuxO and (Cu0.77Ti0.23)Ox are both mainly composed of Cu2O crystallites, their thickness is equal to 430 nm, and their morphology is mostly porous. However, the copper-titanium oxide mixture exhibits higher SR values than the reference sample and it can be assumed that the amorphous titanium oxide improves the hydrogen gas sensing performance of copper oxide based thin films.”
Moreover, authors have not developed a complete hydrogen gas sensing mechanism of (CuTi)Ox thin films, yet. In the literature, gas sensing performance have not been sufficiently investigated and a wide range of research still has to be done. In the near future thin films of various Cu and Ti concentrations will be annealed at 523 K and 573 K and hydrogen gas sensing tests will be performed at operating temperature of 473 K, 523 K and 573 K. In the next step, thin films with certain Cu concentrations, that exhibit the best sensor response, will be chosen for further measurements. For those thin films hysteresis tests and measurements with lower H2 concentrations of e.g. 30 ppm, 50 ppm, 70 ppm will be performed.
Furthermore XPS studies were done not only for as-deposited and annealed thin films, but also for some specific experiments. The authors measured the chemical state of the surface in-situ using XPS before and after exposition of the sample to hydrogen. These measurements allowed for the determination of the change in oxidation state of the investigated thin films. Measurements were made before and after exposure to H2. Measurements were made after exposition for exposure times of 15 minutes and 60 minutes. Those results clearly showed that the surface of the thin films undergoes the reduction process from CuO or mixed CuO-Cu2O to pure Cu2O, which is also a part of the mechanism responsible for changes in electrical resistance shown in the proper graphs. The authors are showing some of the results below (taken directly from the CasaXPS software) only for the reviewer information.
|
a) |
b) |
|
|
|
Results of the XPS measurements of CuxO thin films annealed at 200 °C for Cu2p region:
a) before and b) after exposure to hydrogen
Reviewer:
How were the samples wired for H2 sensing measurement?
Authors:
All measured thin films used for hydrogen gas sensing measurements were deposited on corundum ceramic substrates with interdigitated platinum-gold electrodes. The thin films were deposited only in an active area. The contacts were mechanically masked during the deposition. Below are images of the substrate before the deposition of thin films and figure showing dimentions of the substrate provided by the manufacturer (BVT Technologies).
|
a) |
b) |
|
|
|
Images of: a) the substrate before the deposition and b) figure with the substrate dimentions
Reviewer:
Miller indices are supposed to be labeled when doing XRD analysis.
Authors:
Fig. 2 was edited, each sample was highlighted with different colour. In addition, Miller indices were added.
Figure 2. The diffraction patterns of as-deposited and annealed at 473 K TiOx, (CuTi)Ox, and CuxO thin films.
Reviewer:
In Fig 5a, Why is the resistance of TiOx much higher than the rest of samples?
Authors:
The resistivity of the TiOx sample after thermal treatment measured at room temperature was 3.8·103 Ωcm. The resistivity value obtained in this work falls within the wide range reported in the literature which is 10-4 ÷105 Ωcm [1–4]. Quite high value of the resistivity may be explained by a large amount of the amorphous phase present in the thin film, as evidenced by the low intensity of the peaks visible in the XRD (Figure 2). The resistivity of (CuTi)Ox thin films is lower due to impact of the copper oxides, what states in accordance with other works [5].
- Banakh, O.; Schmid, “P.E.”; Sanjines, R.; Levy, F. Electrical and Optical Properties of TiOx Thin Films Deposited by Reactive Magnetron Sputtering. Surf. Coatings Technol. 2002, 151, 272–275, doi:10.1049/cp:19951163.
- Assim, E.M. Optical Constants of Titanium Monoxide TiO Thin Films. J. Alloys Compd. 2008, 465, 1–7, doi:10.1016/j.jallcom.2007.10.059.
- Grigorov, K.G.; Grigorov, G.I.; Drajeva, L.; Bouchier, D.; Sporken, R.; Caudano, R. Synthesis and Characterization of Conductive Titanium Monoxide Films. Diffusion of Silicon in Titanium Monoxide Films. Vacuum 1998, 51, 153–155, doi:10.1016/S0042-207X(98)00149-3.
- Nguyen, T.T.N.; Chen, Y.H.; He, J.L. Preparation of Inkjet-Printed Titanium Monoxide as p-Type Absorber Layer for Photovoltaic Purposes. Thin Solid Films 2014, 572, 8–14, doi:10.1016/j.tsf.2014.09.054.
- Mazhir, S.N.; Harb, N.H. Influence of Concentration on the Structural, Optical and Electrical Properties of TiO2 : CuO Thin Film Fabricate by PLD. IOSR J. Appl. Phys. 2015, 7, 14–21, doi:10.9790/4861-07621421.
Reviewer:
The authors mentioned that adding an additional layer of catalyst is a common procedure for making a gas sensor. Does the samples tested in this manuscript have a catalyst layer?
Authors:
Thin films described in the manuscript were fabricated without catalyst layer, however in the literature Au [6] and Li [7] were introduced into CuxO-TiOx system as a catalyst to improve the hydrogen gas sensing performance. Compared to other works, thin films presented in the manuscript exhibit relatively low operating temperature (473 K), while an optimal working temperature of 573 K was observed in other works. A systematic study of hydrogen gas sensing properties are planned in the nearest future. CuTiOx will be annealed at higher temperatures to induce copper oxidation to CuO and hydrogen gas sensing properties will be measured at higher working temperatures. For thin film with the best sensing parameters, further studies such as sensor stability test, hysteresis test, sensing in lower H2 concentrations will be performed. Furthermore, Pd catalyst layer will be deposited to compare the gas sensing parameters with and without catalyst layer.
- Barreca, D.; Carraro, G.; Comini, E.; Gasparotto, A.; MacCato, C.; Sada, C.; Sberveglieri, G.; Tondello, E. Novel Synthesis and Gas Sensing Performances of CuO-TiO2 Nanocomposites Functionalized with Au Nanoparticles. J. Phys. Chem. C 2011, 115, 10510–10517, doi:10.1021/jp202449k.
- Torrisi, A.; Ceccio, G.; Cannav, A.; Lavrentiev, V.; Hor, P.; Yatskiv, R.; Vaniš, J.; Grym, J.; Fišer, L.; Hruška, M. Chemiresistors Based on Li-Doped CuO–TiO2 Films. 2021.

Reviewer 3 Report
see attached file

Author Response
The detailed responses are in the attachment (together with figures).
Answers to the report of Reviewer #3
on the manuscript entitled: “Hydrogen gas sensing properties of mixed copper-titanium oxides thin films”
Authors: Ewa Mańkowska, Michał Mazur, Jarosław Domaradzki, Piotr Mazur, Małgorzata Kot, Jan Ingo Flege
Authors:
We would like to express our gratitude for your remarks, which let us improve our manuscript. We have taken them into account in the revised version of our paper.
Answering to the Reviewer’s remarks, we have introduced some revisions in the manuscript.
Reviewer:
More recent references should be added to the introduction, e.g. when talking about oxide examples (line 45-47) the references are very old. In my opinion, there are more recent works that could be included such as doi.org/10.1016/j.snb.2023.133348, doi.org/10.1016/j.ijhydene.2022.01.036, etc.
Authors:
The references in the introduction were replaced by more recent works.
Reviewer:
Line 62 remove the bold writing.
Authors:
The line was unbolded.
Reviewer:
Fig. 2 is a bit confusing in my opinion. If possible improve the presentation perhaps by using different colours for the various samples.
Authors:
Fig. 2 was reedited, each sample was highlighted with different colour. In addition, Miller indicates were added.
Figure 2. The diffraction patterns of as-deposited and annealed at 473 K TiOx, (CuTi)Ox, and CuxO thin films.
Reviewer:
The sensing part should be increased. You present the work as a hydrogen sensor so I think some more data on gas sensing should be included. Like repeatability tests, sensor stability.
Authors:
The repeatability test were performed by repeating three cycles of introducing 200 ppm of H2 and air afterwards. The standard deviation from sensor response (SR) was ~0.05 and was mostly caused by a slight drift of the baseline. However the shape of resistance changes as a function of time of hydrogen flow was similar in each cycle. As an example, in Figure 6 the results of repeatability tests were shown performed for (Cu0.77Ti0.23)Ox thin film. For this thin film the mean SR value and standard deviation was equal to 1.35± 0.04.
“Furthermore, repeatability tests were performed. The performance of thin films were repeatable and small deviations from the sensor response were mainly caused by a slight drift of the baseline. Figure 6 shows the resistance changes in 200 ppm hydrogen and an atmosphere of ambient air for (Cu0.77Ti0.23)Ox thin film. The mean SR value was equal to 1.35 and the standard deviation was 0.04.”
Figure 6. Three cycles of changes of electrical resistance of (Cu0.77Ti0.23)Ox thin films annealed at 473 K upon exposure to 200 ppm of H2.
Reviewer:
Have you performed selectivity tests with other gases?
Authors:
Selectivity tests are still under development, however in this work the main idea was to show the hydrogen gas sensing of (CuTi)Ox thin films.
Reviewer:
- Have you performed hysteresis tests?
- Have you calculated the detection limit (LOD) of your sensors, it should be reported.
Authors:
In the table 1 limits of detection were calculated, however authors have not reported them in the manuscript, as the calculation based on only four hydrogen concentrations are not fully valid. We believe that limit of detection might be lower than values presented in the table. In addition, we have not performed hysteresis tests so far. However, in the near future an extensive works concerning gas sensing performance are planned. It is planned that the work will be carried out within the next year. It is not possible to carry out the measurements and their analysis in time for the responses to the reviewers.
Thin films of various Cu and Ti concentration will be annealed at 523 K and 573 K and hydrogen the gas sensing test will be performed at operating temperature of 473 K, 523 K and 573 K. In the next step, thin films with that Cu at. concentration that exhibiting the best sensor response will be chosen. For those thin films hysteresis tests and measurements with lower H2 concentrations e.g. 30 ppm, 50 ppm, 70 ppm will be performed.
Table 1 Limit of detection of hydrogen for (CuTi)Ox thin films at operating temperature of 473K
|
Thin film |
LOD H2 (ppm) |
|
CuxO |
200 |
|
(Cu0.77Ti0.23)Ox |
120 |
|
(Cu0.56Ti0.44)Ox |
150 |
|
(Cu0.41Ti0.59)Ox |
60 |
|
(Cu0.23Ti0.77)Ox |
- |
|
TiOx |
60 |
Reviewer:
In Figure 5b I would use different colours to highlight the different samples.
Authors:
The Figure 5 was reformatted and each sample was highlighted with different colour. Colours for specific thin films are adequate to colours in Figure 2.
Reviewer:
Have you performed sensor tests in the presence of humidity?
Authors:
“The ambient air flow was provided by Atlas Copco GX3 FF compressor with integrated refrigerant air drier that provides a humidity level <10% and activated carbon adsorbing filter (Donaldson AK series).”
Reviewer:
It would be good to compare your sensor with others reported in the literature. Perhaps by constructing a table or having a discussion on the performance of your sensor compared to others.
Authors:
A short comment of results presented in literature was added.
“Table 2 summarizes the hydrogen gas sensing parameters of copper and titanium oxide nanomaterials available in the literature. To date, hydrogen-sensitive structures operating at 473 K have been mainly manufactured as double layers [32] in which, for example, TiOx was in the form of nanotubes [35]. To obtain a nanocomposite with dispersed oxides, an additional thermal treatment was used [40]. Similarly to this work, the hydrogen gas concentration was in the range between 100 and 1000 ppm. In general, in all these works, the SR does not exceed 2.0 at an operating temperature of 473 K, but the response of the sensor was determined by such a variety of methods that it is difficult to compare individual results with each other. However, based on the literature and the results presented in this study, it seems worth-while to carry out an in-depth study on the sensing properties of (CuTi)Ox thin films. It is worth mentioning that the results presented in this work, in comparison to those presented in the literature, showed thin oxide films with different crystal structure. The phenomena of gas sensing in copper-titanium oxides are mostly investigated in the structures in which copper oxides mostly refers to crystalline structure of pure CuO [41] or Cu2O-CuO structures [32,35,40,64]. In the case of this work, the copper oxides are composed of the nanocrystallites of Cu-Cu2O or Cu2O, while CuO is only observed on the surface. According to the authors best knowledge, such thin films of mixed oxides of copper at various oxidation states with titanium have not been reported to date.”
Table 2. The summary of sensors responses of nanomaterials based on copper and titanium oxides measured at the working temperature of 473 K
|
Heterostructure |
Hydrogen concentration (ppm) |
Sensor Response at 473 K |
Equation of SR |
Reference |
|
(CuTi)Ox thin films |
1000 ppm |
1.86 |
Rgas/Rair |
This work |
|
CuO-TiO2 |
1000 |
2.0 |
Rgas/Rair |
[40] |
|
CuO thin film on TiO2 nanotube |
1000 |
2.0 |
(Igas-Iair)/Iair |
[35] |
|
CuO-TiO2 |
800 |
150% |
(Rgas-Rair)/Rair |
[38] |
|
TiO2/CuxO |
100 |
10% |
(Rgas-Rair)/Rair |
[32] |
|
TiO2/CuxO |
100 |
43% |
Rair/Rgas |
[63] |
Reviewer:
There are some formatting and writing errors please review.
Authors:
The manuscript was reviewed.
Reviewer:
Your sensor works at 473 K what is the behaviour at lower temperatures?
Authors:
Authors performed tests at operating temperature equal to 423 K, the changes of the resistance were evident, however, the SR was mediocre. Nevertheless, sensor responses presented in the work at operating temperature of 473 K is still better than the ones reported in the literature [1–6].
- Barreca, D.; Carraro, G.; Comini, E.; Gasparotto, A.; MacCato, C.; Sada, C.; Sberveglieri, G.; Tondello, E. Novel Synthesis and Gas Sensing Performances of CuO-TiO2 Nanocomposites Functionalized with Au Nanoparticles. J. Phys. Chem. C 2011, 115, 10510–10517, doi:10.1021/jp202449k.
- Lupan, O.; Santos-Carballal, D.; Ababii, N.; Magariu, N.; Hansen, S.; Vahl, A.; Zimoch, L.; Hoppe, M.; Pauporté, T.; Galstyan, V.; et al. TiO2/Cu2O/CuO Multi-Nanolayers as Sensors for H2and Volatile Organic Compounds: An Experimental and Theoretical Investigation. ACS Appl. Mater. Interfaces 2021, 13, 32363–32380, doi:10.1021/acsami.1c04379.
- Alev, O.; Şennik, E.; Öztürk, Z.Z. Improved Gas Sensing Performance of P-Copper Oxide Thin Film/n-TiO2 Nanotubes Heterostructure. J. Alloys Compd. 2018, 749, 221–228, doi:10.1016/j.jallcom.2018.03.268.
- Rydosz, A.; Czapla, A.; Maziarz, W.; Zakrzewska, K.; Brudnik, A. CuO and CuO/TiO2-y Thin-Film Gas Sensors of H2 and NO2. 2018 15th Int. Sci. Conf. Optoelectron. Electron. Sensors, COE 2018 2018, 2016–2019, doi:10.1109/COE.2018.8435156.
- Torrisi, A.; Horák, P.; Vacík, J.; Cannavò, A.; Ceccio, G.; Yatskiv, R.; Kupcik, J.; Fara, J.; Fitl, P.; Vlcek, J.; et al. Preparation of Heterogenous Copper-Titanium Oxides for Chemiresistor Applications. Mater. Today Proc. 2019, 33, 2512–2516, doi:10.1016/j.matpr.2020.05.061.
- Torrisi, A.; Ceccio, G.; Cannav, A.; Lavrentiev, V.; Hor, P.; Yatskiv, R.; Vaniš, J.; Grym, J.; Fišer, L.; Hruška, M. Chemiresistors Based on Li-Doped CuO-TiO2 Films. Chemosensors 2021, 9, 246.

Round 2
Reviewer 2 Report
The revised manuscript could be accepted for publication.
Reviewer 3 Report
The authors answered all the questions posed. I recommend their publication in this journal.